# The broader spectrum of in-context learning

## Abstract

The ability of language models to learn a task from a few examples in context has generated substantial interest. Here, we provide a perspective that situates this type of supervised few-shot learning within a much broader spectrum of meta-learned in-context learning. Indeed, we suggest that *any* distribution of sequences in which context non-trivially decreases loss on subsequent predictions can be interpreted as eliciting a kind of in-context learning. We suggest that this perspective helps to unify the broad set of in-context abilities that language models exhibit—such as adapting to tasks from instructions or role play, or extrapolating time series. This perspective also sheds light on potential roots of in-context learning in lower-level processing of linguistic dependencies (e.g. coreference or parallel structures). Finally, taking this perspective highlights the importance of generalization, which we suggest can be studied along several dimensions: not only the ability to learn something novel, but also flexibility in learning from different presentations, and in applying what is learned. We discuss broader connections to past literature in meta-learning and goal-conditioned agents, and other perspectives on learning and adaptation. We close by suggesting that research on in-context learning should consider this broader spectrum of in-context capabilities and types of generalization.

In recent years, there has been substantial excitement about meta-learning (Finn et al., 2017; Wang, 2021). A notable substream of this work focuses on memory-based meta-learning (Santoro et al., 2016; Wang et al., 2016; Ortega et al., 2019), in which the inner learning process occurs solely within the memory (or activations) of a network. This excitement has been reignited by the recent demonstrations of In-Context Learning (ICL) of tasks from a few examples in transformer language models (Brown et al., 2020; Dong et al., 2022). These in-context learning abilities have been suggested to emerge from particular properties of the training data, such as bursty patterns in the training sequences (Chan et al., 2022a) or repeated parallel structures (Chen et al., 2024).

There has been substantial interest in studying ICL from theoretical and mechanistic perspectives (Xie et al., 2022; Von Oswald et al., 2023; Pan et al., 2023; Akyürek et al., 2022; Wang et al., 2023; Zhang et al., 2024; Swaminathan et al., 2024; Raventós et al., 2024). However, the majority of these works have focused on few-shot supervised ICL—where the goal is to learn an input-output function from a few supervised input-output examples.

Yet language models perform many other kinds of in-context adaptation—such as adapting to tasks from instructions or role-play (Reynolds & McDonell, 2021; Shanahan et al., 2023), extrapolating time series (Gruver et al., 2024), or more active exploration and learning in context (e.g. Coda-Forno et al., 2023; Lampinen et al., 2024)—that do not cleanly fit within the few-shot supervised ICL paradigm. Likewise, there is a much broader literature studying more complex types of meta-learned in-context learning (e.g. Wang et al., 2021; Laskin et al., 2023; Bauer et al., 2023).

Thus, there is a gap between our theoretical and mechanistic understanding of ICL in language models—which tends to focus on few-shot supervised ICL—and the more general capabilities for in-context adaptation that they exhibit in practice. Are these qualitatively different phenomena, or can we understand them as part of an overarching meta-learning process? This paper is intended to bridge this gap, and discuss some of the consequences. See Fig. 1 for an overview of our argument.

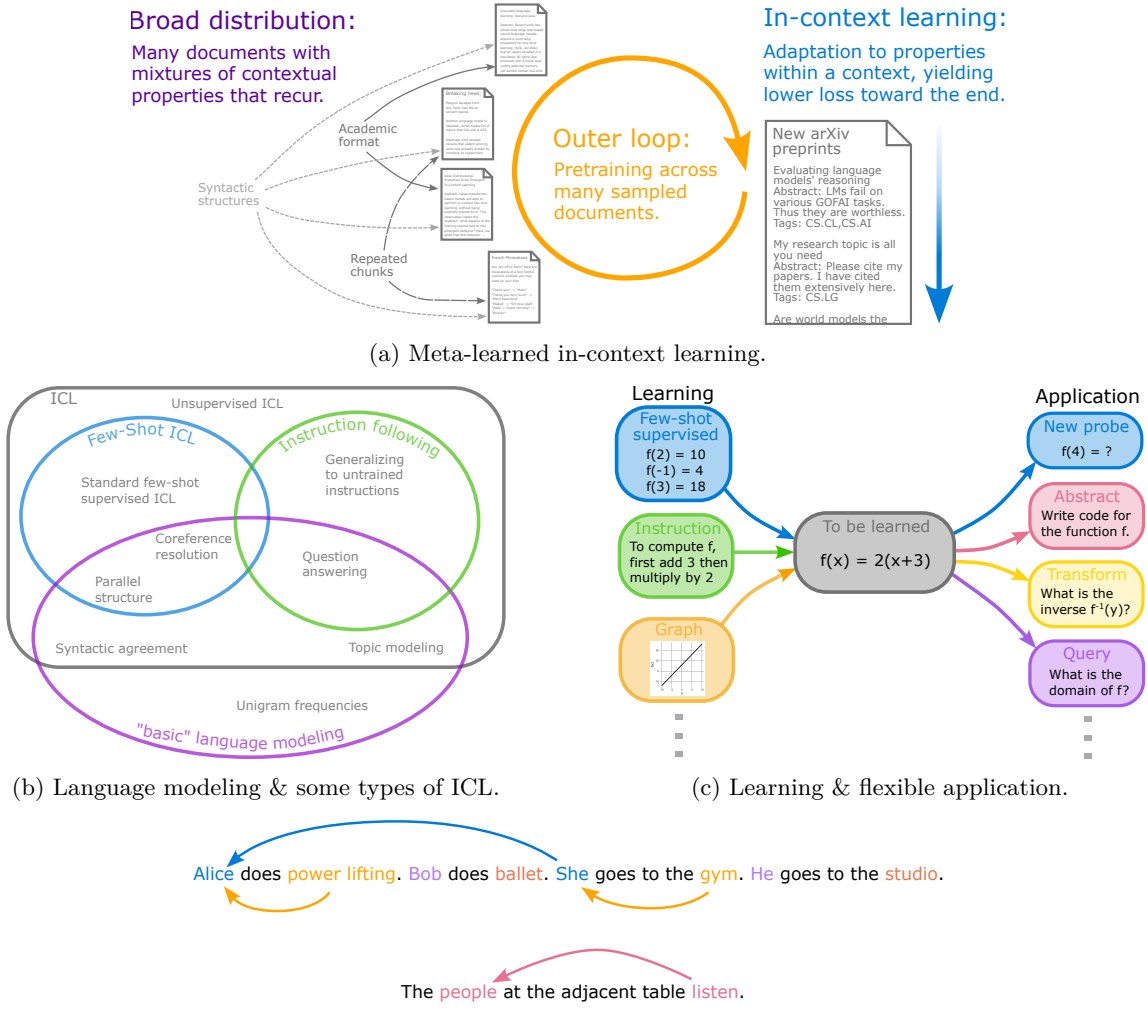

(a) Meta-learned in-context learning.

(b) Language modeling & some types of ICL.

(c) Learning & flexible application.

(d) Examples of in-context learning in language processing.

Figure 1: An overview of our perspective. We see few-shot supervised in-context learning as one part of a much broader space of meta-learned contextual adaptation that spans from basic language capabilities to flexible use of new information. (a) ICL is an "inner-loop" of memory-based adaptation within a context (e.g., a document) that emerges from a meta-level "outer-loop" of learning over a distribution of contexts that mix recurring underlying structures (cf. Brown et al., 2020, Fig. 1.1). (b) This broader view of ICL includes instruction following, and even many aspects of "basic" contextual language modeling. (c) This broader perspective motivates studying ICL (and its generalization) in several components: what is to be learned, how it is learned, and how that learning is applied. In standard supervised ICL, the information to be learned (e.g. a function) is learned from examples, and then applied to new probes. However, the task could be conveyed in other ways, such as a visualization. Similarly, there are many ways the learned information could be applied, e.g. applications to a new domain, or abstractions or transformations thereof. (d) This perspective highlights potential roots of ICL in more basic language modeling. Complex tasks, like binding attributes to entities, matching pronouns with their referents, and then resolving to predictions, parallel the structure of standard few-shot ICL tasks. Even simpler syntactic dependencies are a type of contextual adaptation.

Our goal is to communicate the following perspective:

- There is a broad spectrum of ICL.
- In fact, any sequence task in which context non-trivially reduces loss can be interpreted as eliciting a kind of ICL.
- Standard supervised few-shot ICL is a relatively narrow subset of ICL, in which the contextual dependencies are parallel relations.
- Language models many other types of ICL.

And some corollaries that we draw:

- ICL has potential roots in simpler language processing, e.g. parallelism and coreference.
- Theoretical and mechanistic research in LMs should consider this broader spectrum of ICL.
- For example, there may be important interactions or interference between the mechanisms for different types of ICL.
- Considering generalization is critical when studying ICL.
- Generalization in ICL can be explored in several components: in what is learned, in how it is learned, and in how it is applied. The latter are often under-emphasized.

## 1 Background

We first review some of the literature on meta-learning, grounded agents, and in-context learning in language models.

The few-shot supervised ICL setting fits closely with some of the classic supervised meta-learning settings from earlier work, which focused on learning of classification or regression problems from a few examples, i.e. 'few-shot learning' (e.g. Vinyals et al., 2016; Santoro et al., 2016). This single-input single-output supervised learning setting also fits the few-shot learning demonstrations that first ignited interest in the in-context learning capabilities of language models (Brown et al., 2020) and multimodal models (Alayrac et al., 2022).

However, a wide variety of *other* types of meta-learned in-context learning that have been explored in the meta-learning literature. For example, prior works have explored whether a system can meta-learn to perform in-context reinforcement learning (Wang et al., 2016; Duan et al., 2016). Other works have considered active information-gathering tasks in meta-learning settings where a grounded agent must learn to actively seek out classification labels in an environment in context (Hill et al., 2021); this offers an interesting case study on the broader spectrum of ICL, see Appx. A for some discussion. Other works show an agent can meta-learn to actively experiment to infer some latent causal structure in context, in order to achieve later goals (Dasgupta et al., 2019; Wang et al., 2021; Lampinen et al., 2024). More recent works have scaled to much more complex in-context RL (Laskin et al., 2023), or adapting to complex new multi-agent tasks in context (Bauer et al., 2023). In a different vein, Lake & Baroni (2023) even show that a system can meta-learn to exhibit human-like compositional generalization of languages learned in context. These tasks go far beyond the simple functional input-output mappings of supervised few-shot learning.

There is also a broader area of work (that is *not* typically described as meta-learning) which considers how goal-conditioned agents can perform novel goals at test time. For example, there is a long history of work on language-conditioned agents (e.g. Branavan et al., 2009; Mei et al., 2016; Hill et al., 2020; Lynch & Sermanet, 2021) that demonstrates some generalization to novel instructions. More recently, there has been an interest in various forms of reward/return conditioned policies, as an approach to (offline) RL via conditional imitation (Chen et al., 2021; Srivastava et al., 2019). These works show how training with various types of sequence conditioning can yield generalizable sequential behaviors.

Likewise, language models are not *merely* few-shot supervised learners—they are capable of much more general in-context adaptation from other kinds of in-context cues—such as instructions (e.g. Fig. 1 in Ouyang et al., 2022), explanations (Lampinen et al., 2022), role play (Reynolds & McDonell, 2021; Shanahan et al., 2023), and other kinds of prompting (Schulhoff et al., 2024). They also exhibit more general sequence capabilities (Gruver et al., 2024) and even hierarchical adaptation (Coda-Forno et al., 2023). Thus, the parallels between the types of adaptation considered in the meta-learning and goal-conditioning literature,

and the in-context adaptation capabilities of language models, go far beyond the few-shot supervised ICL regime.

Indeed, some early measures of in-context learning in LMs effectively capture this broader spectrum of in-context adaptation. For example, Kaplan et al. (2020) and later Olsson et al. 2022 measured ICL by the lower loss for tokens later in a document relative to earlier tokens — which incorporates *all* the various contextual structures that support that loss reduction.[1] We adopt a similar perspective in the following section.

## 2 The broader spectrum of ICL: meta-learned contextual adaptation

We interpret in-context learning as the ability to use the context of earlier observations in a sequence task to support predictions (or decisions) later in that sequence, in a way that is meta-learned across a distribution of sequences (cf. Ortega et al., 2019). We define a sequence task to be any task involving making a sequence of predictions/actions ($a_t$) based on a sequence of observations ($o_t$). The relations between observations and actions within and across timesteps evolve according to latent processes. The latent processes for each sequence are sampled from a broader distribution ($\mathcal{D}$) that may have some universally consistent features as well as some that vary. This definition naturally fits Partially-Observable Markov Decision Processes (POMDPs), as well as their common applications in reinforcement learning (Sutton, 2018) and language modeling (Fine et al., 1998). We assume that there is some criterion or loss function ($\mathcal{L}$) by which performance can be measured, such that a lower loss indicates improved performance.

We define a *non-trivial sequence task* to be any sequence task in which context is required for optimal action in a new sequence—that is, one in which the optimal policy ($\pi^*$) that has the context of prior observations can outperform an optimal policy that only can see the current observation and the current time index:[2]

$$\mathbb{E}_{\mathcal{D}} \left[ \mathcal{L} \left( \pi^* \left( o_t, t \right) \right) \right] < \mathbb{E}_{\mathcal{D}} \left[ \mathcal{L} \left( \pi^* \left( o_1, ..., o_t \right) \right) \right]$$

We propose that *any* distribution $\mathcal{D}$ that yields non-trivial sequential dependencies of this type is effectively a meta-learning setting that will give rise to *some* kind of in-context learning capability.

For example, a minimal non-trivial sequence task would be a simple memory task, in which a model is cued with a piece of information at the first step, and then must output that same information after some delay (more complex versions have been used as benchmarks by e.g. Hochreiter & Schmidhuber, 1997). Although this task seems so simple as to hardly count as "learning," a model trained on this task with continuous vector inputs can generalize to store and reproduce vectors at test time that had not been seen in training. In that sense, it exhibits a minimal kind of generalizable in-context learning (of a novel input pattern). This minimal ICL can be enriched with more complex sequential dependency structures (e.g. delayed match-to-sample tasks, injection of noise) and with a few modifications it begins to resemble standard few-shot settings.

From this perspective, the kinds of few-shot supervised in-context learning that have been most studied are merely a consequence of meta-learning of particular types of sequential dependency structures (cf. Chan et al., 2022a), but meta-learned ICL is not restricted to solely dependencies of these types. The full spectrum of meta-learned in-context learning ranges from simple memory, through basic use of context to resolve linguistic dependencies, to supervised few-shot learning, and on to much more complex in-context adaptation. We will explore these themes in the subsequent sections.

---

[1]In this context, it is interesting to reflect on the discussion by Shannon (1951) of how $n$-gram contexts support next-token prediction, and what other (e.g. semantic) structures could likewise constrain predictions in longer contexts.

[2]Comparing to a policy that can see the current time is necessary to rule out trivial sequence tasks such as always producing a particular sequence of outputs—which either requires memory or an appropriate observation that allows inferring the timestep.

# 3 Standard Few-Shot Supervised ICL

How does standard Few-Shot[3] Supervised In-Context Learning (FSS-ICL) of input-output mappings fit into this broader spectrum? Standard FSS-ICL follows a format where a sequence of inputs and outputs are presented, followed by a probe input, for example

> hello->bonjour
> thank you->merci
> goodbye->

This structure offers a particular type of sequential dependencies in which it is the *parallel* relations between inputs and outputs that constrain predictions of the outputs for future inputs.

Specifically, a function $f(x) = y$ offers a *relation* between each input $x$ and its corresponding output $y$. In-context learning of functions thus involves inferring the common relation from pairs of inputs and outputs in context, and applying that relation to infer the corresponding output for a new probe input. The pairs instantiating these relations can either define piecewise function relations (like arbitrary labels for different image classes, e.g. $f(dog) = 1$ and $f(cat) = 2$), or a more unitary relation/function (e.g. $f(x) =$ the capital of country $x$). These features of the function (along with type of input used as a probe) will change the kind of reasoning required, and the likelihood that the model will generalize given its training distribution, but do not change the fundamental nature of the problem: inferring relations and then applying them to a new instance.

Seen from this perspective, supervised ICL resembles certain kinds of analogical reasoning (cf. Gentner, 1983; Holyoak, 2012)—as prior works have noted (Dong et al., 2022). For example, a classic analogy problem like Paris : France :: London : ? requires recognizing the relation on the left (is the capital of), and then applying a higher-order relation (that the relations on each side should be the same) to infer the missing element on the right.

However, there are many other ways that context can inform a model's behavior, without requiring precisely parallel relations; we explore some of these in the next sections.

# 4 Broader demonstrations of ICL in language models

Here, we highlight a few of the many in-context capabilities of language models that do not fit cleanly within the standard FSS-ICL framework.

## 4.1 Task descriptions or instructions

Even base language models (without instruction tuning) adapt behavior to some degree in response to instruction prompts. This capability was used as a baseline in Brown et al. (2020), for example translating from a prompt saying "Translate English to French," and likewise in subsequent instruction-tuning papers (e.g. Ouyang et al., 2022; Wei et al., 2022). In some cases, task descriptions alone yield performance nearly as strong as few-shot prompting, though combining them with few-shot examples often improves further. The ability to adapt to a task from its description is thus a similarly fundamental aspect of in-context learning, that is increasingly important (as fine-tuning mostly focuses on instruction following for ease of downstream use).

## 4.2 Role prompts

Reynolds & McDonell (2021) responded to the earlier few-shot supervised prompting of language models by illustrating the broader set of zero-shot prompting methods for achieving task performance. One effective technique was very simple: presenting the question in a format that indicated the answer came from an

---

[3]Note that recent works on "many-shot ICL" (Agarwal et al., 2024; Anil et al., 2024; Bertsch et al., 2024) have begun to erode the "few-shot" nature of this type of ICL; we use the term here due to its broad adoption following Brown et al. (2020).

expert. For example, for translating from French to English, the authors used the following prefix before the model's generation:

> The masterful French translator flawlessly translates the phrase into English:

and found that it provided performance better than simple few-shot prompts. This illustrates how in-context adaptation need not rely merely on task descriptions or instructions, but can also benefit from other cues that effectively encourage the model to "role-play" a different persona (cf. Shanahan et al., 2023).

### 4.3 Learning from explanations

Lampinen et al. (2022) show that few-shot prompts can be augmented with explanations *after* the answers, and that this can improve model performance beyond instructions or other controls. While explanations do not directly fit into FSS-ICL, it is clear how they might constrain what models infer from ambiguous or difficult examples, and thereby improve performance.

### 4.4 Unsupervised ICL

Agarwal et al. (2024) explore a variety of ICL phenomena, but one of particular interest from our perspective is *unsupervised* ICL—just providing examples of the problems in the prompt, without their solutions—can improve model performance substantially. This benefit does not fit into the classical framework of supervised ICL, because there are no labels to relate to. However, as the authors note, it's possible that for common tasks (e.g. mathematics problem sheets), the mere presence of relevant information may help to cue the skills in question; answers will only be strictly necessary in cases where the task is truly novel. (See also cases where *random* answers in the shots nevertheless yield accurate task performance—the model may often recover tasks from training rather than truly learning something "new" in context (Min et al., 2022).) However, it is possible that unsupervised ICL might work even if the task has not been seen in training, if the training data distribution sufficiently constrains the functions that might be inferred from the examples.

### 4.5 Extrapolating time series

Gruver et al. (2024) find that language models can extrapolate time-series data comparably to domain-specific models. In some cases, the language models appear to accurately integrate multiple components in a time series, such as linear trends combined with seasonal fluctuations. This example illustrates how language models can learn complex contextual dependency structures, even when the boundary between task "inputs" and "outputs" is not clearly defined.

### 4.6 Meta in-context-learning

Coda-Forno et al. (2023) explore what they call "meta-ICL"—when language models learn a sequence of multiple related few-shot tasks in context, language models learn more rapidly at the later tasks than the earlier ones. That is, the authors essentially nest another outer loop of tasks within the context, and find that the models adapt at both levels.

## 5 Roots of ICL in language processing

In the previous section, we highlighted some of the sophisticated forms of in-context learning that language models exhibit. Where do these capabilities originate? Chan et al. (2022a) highlighted how standard few-shot classification ICL could emerge from simple statistical properties—burstiness and long tailed distributions—that are known properties of language. Likewise, Chen et al. (2024) study how parallel structures support learning of ICL. Other works have highlighted how more complex contextual structures in pretraining may contribute to in-context learning (Han et al., 2023). Here, we correspondingly suggest that the broader spectrum of in-context learning capabilities arises analogously from the multi-scale sequential dependency structures in language datasets, and the need to adapt to the long tail of information in the training corpus.

In this section, we illustrate this perspective with examples connecting various kinds of ICL to linguistic properties.

## 5.1 Coreference resolution

Coreference resolution is a long-standing challenge for language models. It consists of making links between multiple ways of referring to the same entity. One common context in which this problem occurs is the use of pronouns in a sentence. Resolving which entity a pronoun refers to requires a basic kind of in-context learning, which can range from simple to complex depending on the context. See Fig. 1d for an example that follows a roughly standard few-shot ICL format.

For a slightly more challenging example, consider the following pair of questions from the Winograd Schema Challenge (Levesque et al., 2012)—a challenging test of coreference resolution:

> The trophy doesn't fit in the brown suitcase because **it**'s too big. What is too big?
>
> The trophy doesn't fit in the brown suitcase because **it**'s too small. What is too small?

Although the sentences are quite structurally similar, the answer to the final question differs in the two cases. In the first case, it is the trophy that is too big; in the second, it is the suitcase that is too small. Resolving the reference in each case requires integrating information from multiple portions of the sentence. Specifically, the sentence structure can be very loosely thought of as presenting an ICL task like the following:

> [A] doesn't fit in [B] (thus bigger = [A] and smaller = [B]). [it] is too big/small. What is [it]?

This rewritten formulation is more similar to classic FSS-ICL, where the context introduces novel entity-property bindings, and the model needs to generalize these bindings to produce the correct output for the probe. Of course, the actual structure of the Winograd-style tasks is different, because the bindings are introduced implicitly, and mediated by semantic knowledge. But this alteration only makes the task more challenging. Thus, models that can effectively do coreference resolution in challenging Winograd tasks exhibit more sophisticated kinds of ICL, that move towards standard few-shot ICL (but remain restricted to a more constrained scope of in-context tasks corresponding to simple reference binding).

Relatively early transformer language models such as GPT-1 and 2 (Radford et al., 2018; 2019) and RoBERTa (Liu et al., 2019) made substantial progress on achieving successful coreference resolution in these challenging settings[4]—thus marking improvements along the continuum to more complex and general ICL.

## 5.2 Parallel structure

Another salient form of ICL in language is *parallel structure* or parallelism (cf. Chen et al., 2024): when similar syntactic structures are used for multiple elements, to emphasize commonalities. For example:

> Alex has a pet snake.
> Blake has a pet hamster.

The similar constructions of the above sentences highlight the common underlying structure. Parallel structure is likely present for both communicative reasons—it can facilitate comprehension (e.g. Frazier et al., 1984; Carlson, 2001)—and lower level ones, such as syntactic priming that causes people to repeat recent structures (Pickering & Branigan, 1999; Mahowald et al., 2016). In any case, these parallel structures are present in the natural language data from which language models learn—indeed, LMs learn to reproduce features like structural repetition (Sinclair et al., 2022)—and thus they may support models learning the kinds of parallel relations used in few-shot supervised ICL.

---

[4]With two caveats: 1) these earlier models were generally evaluated through fine-tuning, as their ICL abilities were not sufficient to learn the schema resolution task in context 2) some, though not all, of their performance may have resulted from simpler associations (e.g. Sakaguchi et al., 2021).

### 5.3 Word-sense disambiguation

Another challenge for earlier language processing models was word-sense disambiguation (Navigli, 2009)—using context to distinguish which of several possible word meanings is intended. For example, the word "bank" can refer to a financial institution or a river's edge, and context is needed to determine which meaning is intended. Like coreference resolution, modern language models have advanced substantially in word-sense disambiguation (Loureiro et al., 2021; Sainz et al., 2023; cf. Lepori et al., 2024).

From our perspective, word-sense disambiguation involves a limited kind of ICL: learning probable topics using earlier context (e.g. phrases like "went to the river"), and then using those to disambiguate the word in context. Of course, this is a more limited kind of learning; each word can only have a few possible meanings. However, this is not so different from classic meta-learning settings where a few labels are repeatedly shuffled and reused for a (relatively) small set of classes during training (e.g. Santoro et al., 2016)—which can nevertheless yield a generalizable ability for certain kinds of in-context learning.

### 5.4 Subject-verb agreement

While the more complex linguistic structures above were challenging for early language models, even the earliest language models exhibited certain simpler kinds of in-context adaptation. For example, the earliest autoregressive neural language models (Elman, 1991)—which used extremely simple architectures—nevertheless exhibited the ability to generalize subject-verb agreement patterns to novel sentences, even with complex dependency structures.

For example, in a sentence like "**Boys** who *Fred chases* **feed** cats" the models would correctly use context to infer that the first noun (boys) should link to a plural verb form, and thus would predict only plural verbs as valid continuations where "feed" occurs, despite the other nested singular noun-verb dependency. Thus, the models had learned over the training corpus how to use the information presented earlier in a novel context (i.e., the subject) to make more accurate predictions later (i.e., the verb conjugation). Thus, gneralizing syntactic dependencies can be seen as a simple example of minimally-generalizable in-context learning.

### 5.5 Topic modeling

While the above examples focus on relatively discrete and structured dependencies, language models accommodate much richer kinds of structure that may elicit more diffuse forms of in-context learning. For example, the models may adapt to an article that seems to be talking about depictions of humans in prehistoric art, and thereby infer that later paragraphs will tend to follow that theme or its nearby neighbors. This kind of contextual adaptation is likely driven in part by more diffuse features—such as the probabilistic relation of words to topics used in classic topic models (e.g Blei & Lafferty, 2006)—rather than the crisp dependencies of subject-verb agreement. Nevertheless, these fuzzier dependencies afford a generalizable kind of meta-learned use of context to make more accurate predictions. This example illustrates how ICL can influence language processing beyond the discrete dependencies above.

## 6 Generalization of ICL

Taking a broader perspective on ICL allows us to see a variety of phenomena falling on its spectrum. One key issue that emerges from this perspective is the need to focus on *generalization*. While behaviors ranging from word-sense disambiguation to mapping images to labels to instruction following can all be interpreted as types of ICL, they will not yield an equally broad spectrum of generalization. A model may exhibit narrow word-sense disambiguation, but not show any generalization to more interesting learning in context. A few-shot prompted language model may generalize well to certain kinds of held-out tasks, but fail to generalize to others that are sufficiently far from (or in conflict with) the training distribution. Thus, in characterizing ICL it is important to study the extent of its generalization.

We suggest that there are several partly-distinct dimensions along which ICL generalization can be considered (Fig. 1c): generalization to learning new information, generalization in how information is learned, and generalization in the subsequent application or evaluation of that learning. While these dimensions cannot

be entirely disentangled, we believe it is conceptually useful to distinguish between them, because each lends itself to different kinds of evaluation approaches.

## 6.1 Learning something novel

The first kind of generalization is generalizing with respect to what can be learned in context. By this, we refer to assessing the model's ability to learn genuinely-novel tasks in context—rather than merely recovering tasks observed in training—and how that ability depends upon the relationship between the new task and the training task distribution.

Many prior works studying the theoretical basis of in-context learning have *already* evaluated some kind of generalization to learning novel tasks in context, within the scope of their paradigms. For example, Xie et al. (2022) considered generalization under distribution shift, and Chan et al. (2022a) evaluated few-shot learning of novel categories. Likewise, theoretical studies of in-context learning of linear & nonlinear regression tasks (e.g. Zhang et al., 2024; Raventós et al., 2024; Frei & Vardi, 2024) have considered how ICL generalizes to learning when the function to regress (or the shot inputs) are sampled from different distributions. However, as our theoretical understanding of in-context learning expands, we hope that studies will consider generalization beyond the FSS-ICL paradigm. Some works have already explored this in some cases, for example, Ramesh et al. (2024) study compositional generalization to function compositions specified through explicit instructions rather than through examples. However, it is also interesting to consider generalization *across* different methods by which information can be presented.

## 6.2 Learning in varied formats

As we have outlined above, language models can learn the same task in context from many different types of cues. It is therefore interesting to understand the extent to which models perform similarly or differently across different ways of presenting the task. There has been some investigation of models *variability* in FSS-ICL performance from minor changes to the format or ordering of the prompt examples (e.g. Lu et al., 2022; Sclar et al., 2023). However, we believe there should be more systematic investigation of other types of ICL. We show some preliminary examples of such analyses in Appendix C—showing similarities and differences in how a language model adapts to tasks from prompts containing instructions, explanations and examples.

In models trained on broader distributions, we think it will be particularly interesting to characterize the interaction *between* different forms of in-context learning—for example, between learning from examples and instructions, or with visualizations in a vision-language model—which has not been studied as thoroughly (though cf. Liu et al., 2024). For example, to what extent does learning a novel task from examples benefit from training experiences with performing related tasks from instructions? More generally, a model trained with a broad distribution of task presentation strategies might generalize to learning tasks presented in new ways.

However, we suggest that it is equally interesting to study how flexibly models can apply information learned in context.

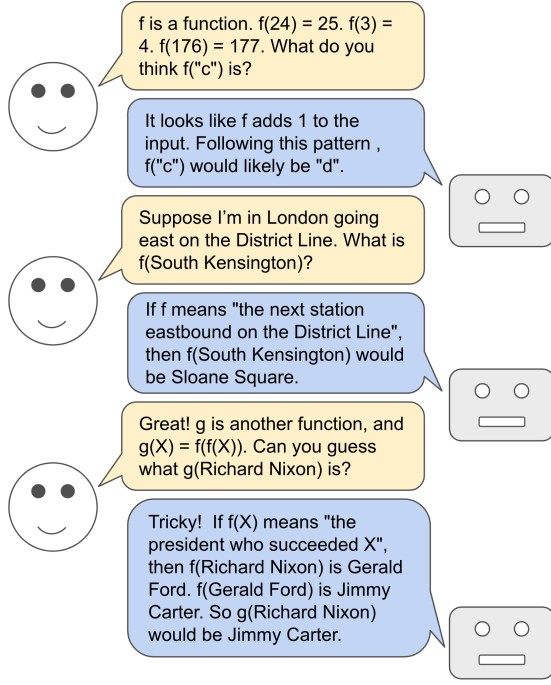

Figure 2: Examples of learning a function from examples in context, and then using that knowledge more flexibly than the initial examples would suggest. (Edited from a real interaction with an LM; the full transcript is in Appx. B.)

### 6.3 Flexibly applying what is learned

The final type of generalization occurs in applying what is learned in context. For example, if a model learns a function over integers, could it apply it correctly to fractions? If there is a reasonable analogical mapping, could the model generalize to apply the same function relation to letters, or to subway stops or presidents? Indeed, in simple cases models can (Fig. 2; Appx. B).

Can the model go beyond applying what it has learned to state abstractions about it, e.g. code implementing the function or explanations of what it computes? There have been a few works looking at versions of this problem—e.g., inferring instructions for a task given examples (Honovich et al., 2023; Liu et al., 2024),[5], or rationales for answers if the prompt contains them (Marasović et al., 2022)—but it has been explored less comprehensively.

We believe that these aspects of flexible reuse and abstraction of what has been learned in context deserve further investigation from empirical, theoretical and mechanistic perspectives. Flexible reuse also illustrates some of the more complex interactions between mechanisms of in-context learning that occur in more complicated problems—for example, flexible application generally requires in-context use of instructions, even if the original information was conveyed through examples.

## 7 Discussion

In this paper, we have outlined a perspective that describes in-context learning as a broad spectrum of related capabilities for adapting behavior to context. From this perspective, any non-trivial sequence task involves some degree of in-context learning. This perspective helps to connect few-shot in-context learning to instruction following and other forms of contextual adaptation, including more basic language modeling capabilities. It also emphasizes the importance of considering generalization in in-context learning—in terms of what is learned, how it is learned, and how it is applied. Here, we discuss some of the potential implications and consequences for research that motivated us to articulate this perspective.

**Beyond FSS-ICL:** Research on ICL should not fixate on the FSS-ICL setting, especially as model deployments are moving to a focus on instruction following. Instead, we recommend exploring the broader spectrum of ICL, and the potential interactions between different types.

**Transfer and/or shared circuitry between different kinds of ICL:** One intriguing possibility is that there may be transfer or shared circuitry between the way models implement different kinds of ICL—analogously to how language models may learn more complex syntactic structures by piecing together what they learn from related simpler ones (Misra & Mahowald, 2024). For example, whether a function is learned from examples or instructions, the application behavior is the same—and thus the circuitry for applying what is learned in context might overlap for the two cases. Similarly, copying circuitry used for more sophisticated ICL (e.g. in induction heads, Olsson et al., 2022) might be learned in part from predicting simpler single-token repetitions, or parallel syntactic structures. In turn, these linguistic structures, and the circuits that support them, might also contribute to more complex kinds of ICL. For example, the use of parallel relations in syntactic structures or FSS-ICL might relate to model performance on analogy tasks (Webb et al., 2023);[6] or more complex types of ICL like meta-learning in context (Coda-Forno et al., 2023). Conversely, if models can use multiple distinct strategies for an ICL behavior (cf. Bhattamishra et al., 2024), then the underlying mechanisms might interact in complex ways. Indeed, several recent works show complex patterns of cooperation and competition between different types of ICL in overlapping circuits (Singh et al., 2025), and transitions between different mechanistic implementations of ICL over the course of training (Park et al., 2024; Yin & Steinhardt, 2025). Considering these interactions would therefore be important to understanding model behaviors and internal mechanisms more completely.

**Interference between different kinds of ICL:** Taking a broader perspective on ICL may provide a useful perspective on some instances where models *fail* to learn the intended task in-context. For example,

---

[5]See also work on "out of context" inferences (Treutlein et al., 2024), though as the name suggests it was not applied to in-context learning.

[6]Though cf. Lewis & Mitchell (2024) and Webb et al. (2024) for some further debate.

this includes e.g. ignoring flipped labels (Min et al., 2022; Wei et al., 2023) or instructions (Webson & Pavlick, 2022), or overly-fixating on label patterns (Yan et al., 2024), or failing to correctly integrate across interactions in context (Qiu et al., 2024). These failures may be due to interference from other kinds of in-context or in-weights learning mechanisms (e.g. more basic associations of labels to particular tasks). Likewise, effects of seemingly-irrelevant factors like prompt formatting or ordering (e.g. Sclar et al., 2023; Lu et al., 2022) may stem in part from models learning about how these features predict task contexts across the training distribution. There may also be dynamic effects over training, in which different types of ICL dominate at different times (Singh et al., 2024; Park et al., 2024).

**Generalizable ICL is not guaranteed:** As the points above highlight, there is no guarantee that models *will* learn some particular kind of generalizable in-context learning just because they are trained on non-trivial sequence tasks; that depends on how the data structure interacts with the model's inductive biases and the target task distribution. As a simple example, *n*-gram language models or bag-of-words models will learn some basic kinds of in-context adaptation, but will not exhibit particularly interesting generalization. More practically, RNNs may be less biased towards certain kinds of ICL than Transformers (e.g. Chan et al., 2022a). Model inductive biases—and whether models are optimized sufficiently to actually learn the task structure (cf. Power et al., 2022)—can shape the kinds of ICL models acquire in practice.

Similarly, where shortcut features (Geirhos et al., 2020) exist, models may learn to use these rather than relying on more complex ICL structures. If models can simply memorize instead of learning to adapt, they often fail to acquire generalizable inner-loop learning (cf. Yin et al., 2020). Likewise, if language models that are trained on purely offline data, there may be causal obstacles to generalizing ICL appropriately in active deployment settings (Ortega et al., 2019; Zečević et al., 2023; though cf. Lampinen et al. 2024). Various properties of the training data may limit the generality of ICL in practice.

**Is in-context learning really learning at all?** Our broad definition of ICL may seem to render the term vacuous—and reinforce broader questions about whether models really "learn" in context at all. While we are sympathetic, we think that interpreting ICL as a type of learning is compatible with the broader meta-learning literature. We hope that most readers will agree that if a model is presented with a truly-novel definition in a test sequence, and uses that to correctly answer subsequent questions in context, it is meaningfully demonstrating learning. From this, it seems a small step to the idea that a model that uses the kind of parallel structures and coreference depicted in Fig. 1d (top) is similarly learning something in context in order to accurately predict later words (as long as the entities are novel). We believe considering the links between these simpler and more complex level of sequential dependencies is motivated by the potential interference or transfer across them.

**Is in-context learning different from other forms of learning?** Our perspective draws somewhat-arbitrary boundaries between timescales: of observations, sequences, and contexts. First, we distinguished learning within and across distinct "observations"—which means that tokenization, for example, may affect whether a task involves "in-context learning" or not. On the other extreme, recent strategies like test-time training (e.g. Rannen-Triki et al., 2024; Akyürek et al., 2024) have begun to remove some distinctions between gradient-based training across contexts and in-context learning within them.[7] In both cases, we believe that these distinctions that isolate in-context learning from shorter- or longer-timescales are partly an artifact of our current approaches to model design, training and evaluation.

An alternative approach would be to see the system as continually adapting to an ever-evolving data stream—a form of continual lifelong learning (cf. Parisi et al., 2019; Abel et al., 2024)—without clear distinctions between "present context" and what is "past data." With this more general perspective, the different scopes of learning would simply become different scopes of dependencies within the same general learning process.

However, it is important to emphasize that when different kinds of learning are implemented through distinct mechanisms, as they are in present models, they may correspondingly have distinct inductive biases. Indeed, transformers generalize differently from in-context and in-weights learning (e.g. Chan et al., 2022b). Likewise, the differences between within- and across-token learning likely drive some of the impact of tokenization decisions on model performance (Singh & Strouse, 2024). Thus, while we believe that the distinctions between

---

[7]As does the common practice of training language models with sequence packing, without masking attention between documents (e.g. Brown et al., 2020).

timescales are somewhat arbitrary from an abstract perspective, the choices we make in distinguishing them in our model implementations has real impacts on learning and generalization.

**Possible connections to natural intelligence:** Statistical learning of patterns in sequential input data are also thought to be important in human language learning (e.g. Saffran et al., 1996). More abstractly, context-sensitive computation is thought to be an important aspect of natural intelligence (Butz et al., 2024). It is therefore interesting to ask whether the kind of multi-scale contextual dependency structures we have highlighted here—in language and beyond—also support the development of capabilities for efficient learning and adaptation in natural intelligence.

# 8 Conclusions

In this paper, we have provided a perspective on in-context learning that situates the few-shot in-context learning in large language models within a broader spectrum of in-context learning. Indeed, we suggest that *any* nontrivial sequential dependencies effectively induce *some* kind of in-context learning. This perspective helps to connect standard supervised ICL to the broader contextual capabilities of language models, such as instruction following or role playing. Our perspective also highlights potential roots of ICL in more basic contextual language processing. Finally, seeing the broader spectrum of ICL suggests several types of generalization that can be evaluated: generalization of what is learned in context, and how flexibly it can be learned and applied. We hope that our perspective will prove useful for researchers interested in the capabilities of large language models, as well as those more generally interested in the links between meta-learning, ICL, goal-conditioned agents, and other research on adaptive sequential behavior.

**A call to action for ICL research:** Our main goal in articulating this perspective is to advocate for ICL research to expand its focus beyond the few-shot supervised setting, by incorporating other kinds of in-context learning and generalization. We suggest that there will likely be mechanistic and behavioral interactions among the many kinds of ICL. Considering these interactions will be necessary to fully understand the generalization behavior and internal functions of large models trained on rich sequential data.

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

## A A grounded case study on the broader spectrum of ICL

For this case study, we draw on an example from earlier work on in-context/meta-learning (Hill et al., 2021). Hill and colleagues considered an in-context language learning task for an embodied agent (depicted in Fig. 3). The agent first had to explore a room and look at each object, as it encountered each it would get a cue describing the name of the object (randomly generated on each episode). Once the agent had encountered all objects (or sufficient time had elapsed), it would get teleported to a new room, where the objects were shuffled, and told to go to a particular object via its arbitrary name.

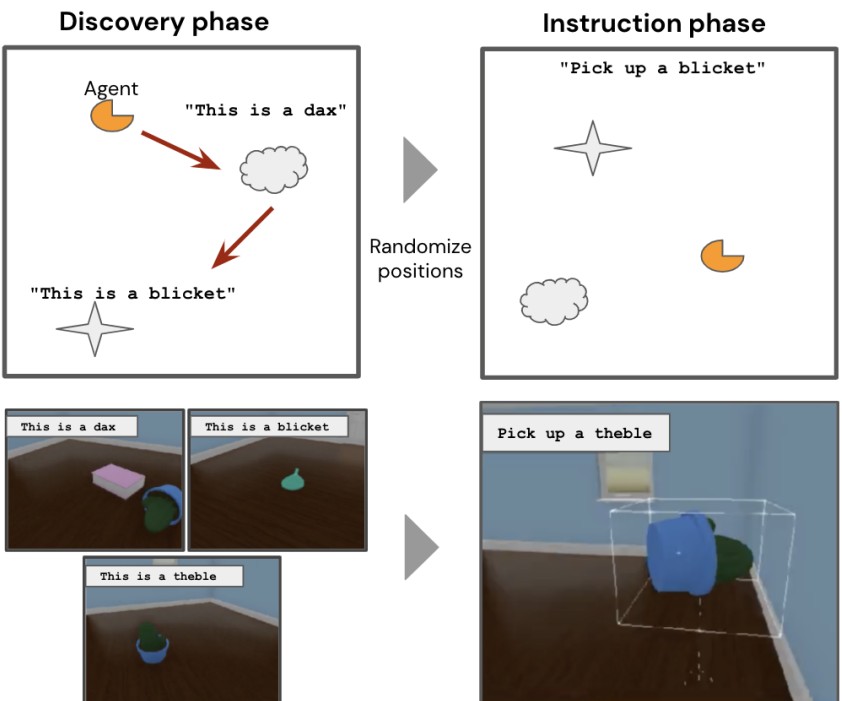

Figure 3: The grounded in-context learning task from Hill et al. (2021). (Figure reproduced from the original paper.)

Thus, the task involves learning in-context the relations between entities and their labels, and then applying it for a probe. In this sense, it is quite similar to the standard supervised few-shot learning tasks described above. However, it has a few notable differences:

1. The mapping changes direction between the context and probe phases. In a standard few-shot categorization task the agent is always given exemplars and outputs labels. The first phase of the Hill et al. task is similar to this standard setting, in that the agent is exposed to a category-label mapping. However, the second phase reverses the direction of this mapping—the agent is given a label, and has to choose the corresponding category. This still involves a kind of parallel relations of course, just one where the directions are reversed between the phases.

2. More fundamentally, however, this task only looks like a standard few-shot categorization task at a high-level of abstraction. From the actual perspective of the model's inputs (visual observations and language tokens) and outputs (low-level control actions), there is *no* simple relation between how the inputs produce outputs in the two different task phases, since the objects and the room change between exposure and test. The structure only appears when abstracting away all the details of perceiving and navigating a 3D environment.

This task already hints that there is nothing particularly special about the structure of the standard few-shot learning setup relative to more general in-context learning. Moreover, we can consider other variations of this task that expose the broader range of ways that a model can learn and apply its knowledge in-context, e.g.:

**Purely linguistic presentation:** Of course, the exposure phase could simply use language to present the object-label relation directly, (e.g. "A book is a dax, ..."). This is still parallel relations, but would require making the more abstract mapping from relations purely in language to the grounded execution of the test instruction with visual inputs.

**Learning from negations:** The names could instead be presented negatively: as the agent approached each object it could be told e.g. "this is neither a dax nor a blicket" and it would have to infer by exclusion what the correct label for each one was.

**Learning from abstract representations:** The wall of the room could have a map with labels on the locations where the objects appear; the agent would have to learn the labels by mapping the space/label relations on the map to those in the room, and then apply its knowledge of those abstract labels in the test phase (after the room and the object locations change).

**Composing knowledge at the test phase:** Hill et al. (2021) also considered several other tests of flexibility, for example one in which the test task is to "put a X on the bed/tray"—thus requiring composing the labels learned in context with other linguistic structures, and following those structures to manipulate objects and their relations to achieve the desired goal.

**Composing multiple learned labels:** The agent could be exposed to mappings like dax = red and blicket = cube in the training phase, and then given a probe task like "find the daxet"—it would then need to infer that "daxet" is a portmanteau of dax and blicket and correspondingly choose the red cube.

**Continual exposure:** The agent could be placed in a series of rooms with different objects; some familiar and some unfamiliar: in each it could both be exposed to new knowledge and tested on prior knowledge. There need be no clear separation between the exposure and test phases.

Each of these variations changes the exact structure of the task the agent is performing, and how readily that structure maps on to the simple setting of standard FSS-ICL. However they do not fundamentally change the nature of the task: gaining information about object-label pairings in the exposure phase(s) which changes in each new context, and then applying those object-label pairings in the probe phase(s) to achieve some instructed goal. That is, across all of these tasks the information earlier in the sequence of experience helps to predict the correct outputs later—even though the exact nature of those dependencies differs dramatically across the tasks.

This case study illustrates how standard few-shot categorization ICL can readily be transformed into many other types of ICL by altering the dependency structures within the sequence, without fundamentally changing the information being learned or tested in the (meta-)learning problem. Furthermore, the task here is just one paper from a broader area of work studying goal-conditioned agents, much of which can be seen as implementing some form of generalizable in-context learning from our perspective.

# B    Flexible reuse chat transcript

Below, we include the full chat for the flexible reuse example, showing how the model (Gemimi Pro; Team et al., 2023) can apply a learned function to quite different instances than the in-context examples. Of course, the successor function is an extremely common function, and thus this should not be taken as evidence of the model actually learning a truly-novel task in context and then flexibly applying it. Rather, this example is merely an illustration of the general idea of flexible reuse.

```
User: f is a function. Here are some examples of its application. f(24) = 25. f(3) = 4. f(176) = 177.
    What do you think f("c") is?
```

```
Gemini: It looks like the function f simply adds 1 to the input. Following this pattern, f("c") would
    likely be "d".
```

```
User: Suppose f(X) = "p". What do you think X is?
```

```
Gemini: If f(X) = "p", and we assume the function operates on the pattern we observed before (adding 1 to
     the input), we need to find the character that comes before "p" in the alphabet. Therefore, X would
    be "o".
```

```
User: Suppose I'm in London and I'm going east on the District Line. What do you think f(South Kensington
    ) is?
```

```
Gemini: This is a fun thought experiment!  It seems like we're now applying the function f to the London
    Underground. If f means "the next station eastbound on the District Line", then f(South Kensington)
    would be Sloane Square.
```

```
User: Great! g is another function, and g(X) = f(f(X). Going west on the Central Line, what is g(
    Liverpool Street)?
```

```
Gemini: Start with the inner function: f(Liverpool Street) means "the next station westbound on the
    Central Line from Liverpool Street".  Looking at a map, that's Bethnal Green. Apply the outer
    function: Now we have g(Liverpool Street) = f(Bethnal Green).  Again, this means "the next station
    westbound on the Central Line from Bethnal Green".
```

```
User: Nice try. But look again, thinking about directions.
```

```
Gemini: Ah, you're right! I missed a crucial detail. We're going westbound on the Central Line. So, f(
    Liverpool Street) is actually Bank. Then, f(Bank) going westbound is St. Paul's. Therefore, g(
    Liverpool Street) = St. Paul's.
```

```
User: Can you guess what g(Richard Nixon) is?
```

```
Gemini: This is a tricky one! We need to figure out what "rules" apply to this new version of the
    function g. Let's consider a few possibilities:
Presidential Succession: If f(X) means "the president who succeeded X", then f(Richard Nixon) would be
    Gerald Ford. Applying f again, f(Gerald Ford) would be Jimmy Carter.  So in this case, g(Richard
    Nixon) would be Jimmy Carter.
```

## C Different types of ICL yield different solutions

In this section, we perform some minimal experiments that illustrate some of our points on the differences and interactions between different types of ICL. Our goal was to compare how in-context learning from instructions, examples, and explanations produces adaptation to the task.

To do so, we adopt the approach of error analysis. Rather than focusing primarily on performance across different prompt conditions, we focus on whether different prompt conditions agree on the response to each question. The motivation for this is that similar learning algorithms should produce more similar patterns of responses—for example, getting the same questions right, and producing similar errors when they are wrong. We thus analyzed how often responses to a question agree for models given different instances of the same prompt type (for example, different sets of instructions for the same task; or different sets of examples in a few-shot prompt), compared to how often responses agree across different prompt types (e.g. instructions vs. few-shot examples).

We use a dataset released in a prior work (Lampinen et al., 2022) that consists of multiple sets of instructions, prompt examples, and explanations of those examples, for a variety of BIGBench (Srivastava et al., 2022) tasks. In particular, we choose 6 challenging tasks which have more than one wrong answer (to yield more signal on response patterns, e.g. if different prompts yield different errors): `crash_blossom`, `crass_ai`, `evaluating_information_essentiality`, `goal_step_wikihow/goal_inference`, `penguins_in_a_table`, `truthful_qa`

We evaluate Gemini 2.0 Flash (Team et al., 2023) on the tasks in question using all combinations of including or not including instructions, examples, and explanations (if examples are present). We avoid using chain of thought, so that performance will not be at ceiling (as we need errors to analyze difference between the in-context algorithms). We analyze the data using mixed-effects logistic regressions that account for the random effects of task and question (e.g. the fact that some questions within a task may be easier than others, which will produce more agreement on the right answer).

We find high similarity in responses overall, mostly due to the fact that the model achieves high accuracy across conditions. However, we do find a variety of interesting patterns of differences that are in keeping with our hypotheses.

**Instructions vs. few-shot examples:** We first focus on comparing prompts containing only instructions to prompts containing only few-shot examples (Fig. 4). We again find that response patterns are significantly more similar between prompts of the same type than prompts of different types ($z = 24.53, p < 10^{-5}$. These results support the claim that instructions and examples specfically induce different modes of adapting to a task, that produce patterns of responses that differ more than the response produced with e.g. different sets of few-shot examples.

**Explanations yield increased similarity:** Finally, we compare prompts containing post-answer explanations for each example to prompts without explanations, with the hypothesis that explanations will result in more consistent patterns of responses by improving task inference. We indeed find that explanations result in more consistent patterns of response (Fig. 5)—thus showing how different modes of in-context learning can interact.

**Comparing prompt types more generally:** We then consider all pairs of prompt types, and compare the similarity within the same prompt type to the similarities between prompt types (Fig. 6; note that we omit empty prompts that lack any task cue from this analysis, as the comparison would be fairly trivial—though interestingly the model still performs well above chance with empty prompts, due to the cues in the question itself). Overall, we find that, as hypothesized, similarity is significantly higher within the same prompt type than between ($z = 24.61, p < 10^{-5}$—thus illustrating that different types of prompts produce different modes of in-context adaptation.

**Full results:** For interested readers, the full similarity structure is visualized in Fig. 7.

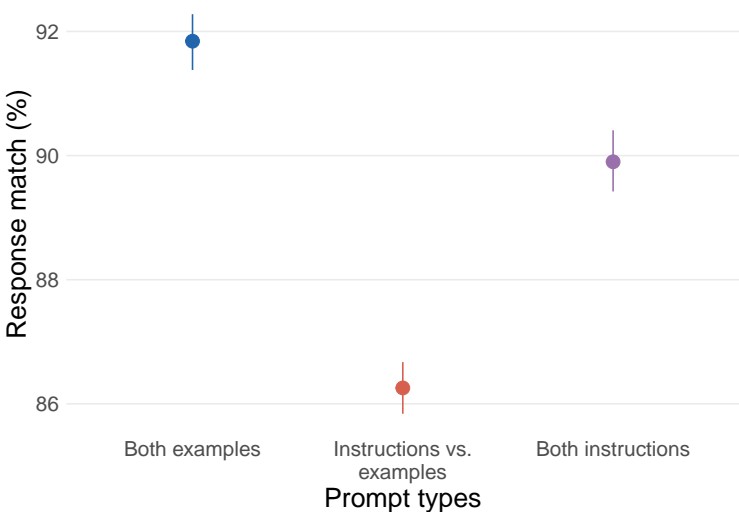

Figure 4: Responses match more between prompts with the same type of ICL (both examples, or both instructions) then between different types (instructions vs. examples).

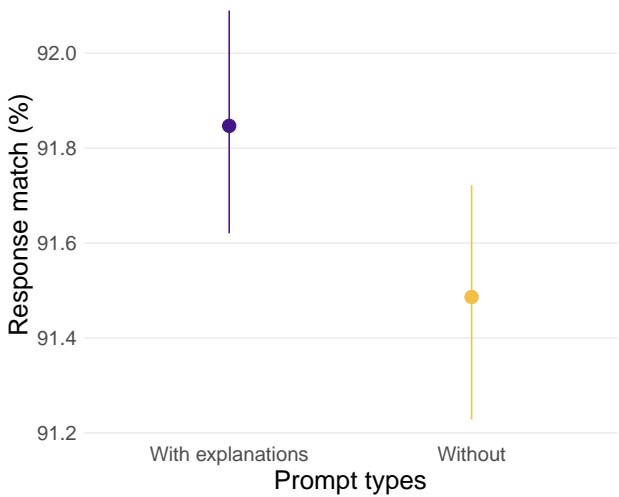

Figure 5: Responses match more between different few-shot prompts that include explanations than they do between different few-shot prompts without explanations.

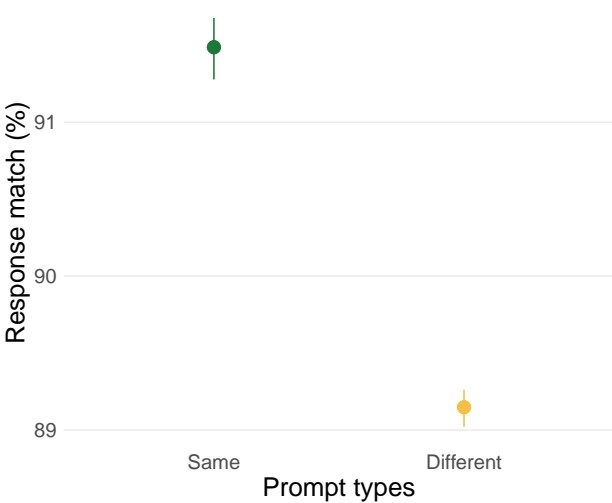

Figure 6: Responses match more between prompts of the same type than between prompts of different types.

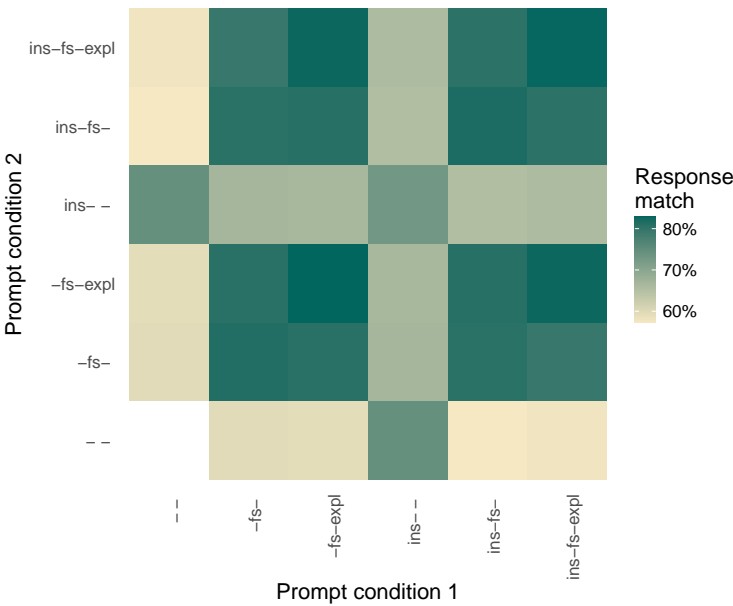

Figure 7: All pairwise similarities in response patterns between the language model when given prompts of different types.

