# OpenReview forum: "The broader spectrum of in-context learning"
_TMLR — Rejected by TMLR_

### Review · Reviewer_Qfvx · 2025-04-03

**Summary Of Contributions:**

The paper argues that ICL extends far beyond the typical few-shot supervised paradigm. Rather than restricting ICL to learning from a few input-output examples via prompt, the authors suggest a “broader spectrum” in which any context-based "learning paradigm" (reduction in loss due to prior context) could effectively qualifies as ICL.

As the main argument, this includes paradigms ranging from basic linguistic dependencies (like coreference resolution or subject-verb agreement) to more complex skills like instruction following, role prompting, time-series extrapolation, and meta-adaptation. They stress evaluating how models generalize in their “inner” ICL loop (similar to a collection of short term memory + implicit memory, if I understand correctly), and call for deeper explorations of interactions between these different forms of adaptation in language models.

**Audience:**

Yes

**Broader Impact Concerns:**

No ethical concerns whatsoever.

**Claims And Evidence:**

No

**Requested Changes:**

## **Experiments**

* More Unified Empirical Demonstrations: Present a single language model evaluated systematically across many “types” of ICL—standard few-shot, instruction-based, role prompts, etc.—to illustrate the paper’s thesis.

* One type of non treditional ICL task with 2-3 diffrent LLMs would be very beneficial, aspeccilly where one would have reasoning /COT mechanism. It would be great to see if such model need reformulation of ICL.

## Connection to other forms of learning (question)
* Prompt tuining is an explicit learning that encode the knowledge in the context, would you count it as ICL as well?


## Writing

* Too many (11) examples for the intrest in study ICL in the intro.

* Repetative terms in some sentences, for example "there is a much broader meta-learning literature studying much more complex types of meta learning"

* Figure 2 is giant, consider shrinking or modify it, also it can be similar to appendix B but a bit bigger text, as now it seems adapted to mobile or two column format.

* Figure 3 on the other hand is a bit small and could be ordered better.

* Transcript: would be easier to read if user had one color and gemini another.

**Strengths And Weaknesses:**

## Strengths
* **Unified Perspective**: First to my knowledge to connect many disparate language model behaviors (e.g., instruction-following, role-prompting, parallel structures) under the same (meta) ICL paradigm name.

* **Clarity About Generalization**: Emphasizes evaluating not just whether a model can do few-shot tasks, but how and where it applies new context-derived knowledge.

* **Relevance to Broader Community**: Bridges ICL with classic meta-learning and goal-conditioned agent research, inviting fresh angles for empirical and theoretical work.

##Weaknesses

* **Too Broad Definition**: The claim that “any sequence task with a context that lowers loss is a form of ICL” might dilute the concept of “learning,” requiring careful distinctions between truly novel learning and simply next-token predictions within the same domain. Arguably, if the model observed similar data, you could say that ICL in the broader aspect is direct to previously known data, and this is indeed a form of learning, from the context, however, I don't see what it falls into. Otherwise, I'm not sure that loss does not decreese on one example can count as non- learning paradigm.

Also, the unexplained phenomenon of loss reduction by changing the format of examples in the traditional few shot supervision ICL shown to mather, so if one could define a rank for stronger and weaker ICL by the loss, it would be great if you could be able to explain such phenomenons within the definitions in this paper.

* **Empirical Results**: The Most examples are conceptual or drawn from past papers. Direct experiments illustrating these “different spectrums” in the same model are not deeply showcased.

I would like to see varying types of ICL on the same model, but also several models for the same type of ICL. as for your definition of ICL, one could claim that the paradigm is learning but QWen do not learn from it, Llama2 does less compared to Gemini2.5 and so on..

p.s.1 Great examples.
p.s.2 Did you write the breaking news by yourself? It was hard to read, but I truly appreciate the humor.

---

> ### Author Response · Authors · 2025-05-02
>
> We thank the reviewer for their thoughtful comments; we are glad they appreciated the perspective (and the humor)
>
> **Goal:**
> Our goal in this paper was primarily to write a survey article highlighting these connections between different types of meta-learning — indeed, TMLR accepts survey articles as long as they draw new connections or propose new problems. We see our main contribution as drawing out these common themes from prior work, emphasizing their connections, and providing a unifying perspective along with suggestions for future research, rather than performing our own experiments. However, as several reviewers requested it, we have added some minimal experiments to the supplement that illustrate some of the themes we raise and address some of the specific requests (see below).
>
> **Definition:**
> * Although our definition is broad, it is based upon the prior literature. It does incorporate several important restrictions restrictions on the breadth of ICL: first, that the loss reduction must be present in expectation over the data distribution (thus, loss failing to reduce on a single example would not rule out ICL), and second that it must be non-trivial (i.e. better than the distribution conditioned on the timestep alone; see footnote 2 for an example of a type of task excluded by this restriction).
> * The notion of measuring in-context learning through the reduction in loss on later tokens was already established in seminal works on scaling laws and induction heads (Kaplan et al., 2020; Olsson et al., 2022). This perspective is also supported by earlier work in the meta-learning literature (Ortega et al., 2019). We therefore believe part of the value of our work is highlighting this general notion of contextual adaptation, and the many types of contextual structures that contribute to loss reduction, more explicitly — and linking them to the prior meta-learning literature — in contrast to the typical assumptions about few-shot in-context learning.
> * If the reviewer has other specific suggestions on revising the definition, while still accommodating the many ways that models can adapt in context (e.g. from instructions alone, rather than just examples), and the breadth of ways this in-context learning can be applied (e.g. producing code for a task than performing a task directly), we would welcome them.
>
> **Supplemental experiment:**
> We have added some brief experiments to the supplement showing comparisons and interactions between different types of learning. In particular, we compare patterns of responses produced by a language model within and across different types of prompts (instructions, few-shot examples, and examples with explanations). We find several interesting and statistically significant patterns: 1) different prompts of the same type produce significantly more similar patterns of responses than responses of different types, 2) specifically, different instruction prompts and different few-shot prompts produce response patterns that are more similar to other prompts of the same type than prompts of the other type, 3) explanations added after each few-shot example increase the consistency of response patterns among few-shot prompts. These results support our claims that different modes of in-context learning yield different modes of adaptation to a task, and that these can interact in complex ways. In our view, these results strengthen our argument for creating a broad unifying framework that highlights the similarities, differences, and interactions among different kinds of contextual adaptation.
>
> **Adjustments:**
> We have reduced citations in the intro, adjusted the figures, added highlighting to the transcript, and made the other minor edits suggested.

---

> > ### Comment · Reviewer_Qfvx · 2025-05-14
> >
> > Thanks for adding the experiments and adressing my concerns.
> > After reading the comment and revision, I am satisfied with the revised version.
> >
> > As a minor friendly note, you make a large use of the term "much" or "much more", that in my opinion in unnececery.

---

> > > ### Author Response · Authors · 2025-05-14
> > >
> > > Thank you for the update! And thanks for pointing out the overuse of "much" — we will cut it down.

---

### Review · Reviewer_4Bs5 · 2025-04-10

**Summary Of Contributions:**

1. The paper proposes a broad conceptualization of in-context learning (ICL). The authors propose that any sequence task where context reduces prediction loss qualifies as ICL, thus extending the notion from few-shot supervised learning to include instruction-following, role-play, and even basic language processing tasks.

2. The work highlights generalization along three dimensions: (1) what is learned in context, (2) how the learning occurs, and (3) how it is applied, spanning the spectrum from simple memorization to genuine novel task learning.

**Audience:**

Yes

**Claims And Evidence:**

Yes

**Requested Changes:**

1. Missing references:
- Understanding In-Context Learning via Supportive Pretraining Data. ACL 2023.
- Understanding In-Context Learning from Repetitions. ICLR 2024.
- Label Words are Anchors: An Information Flow Perspective for Understanding In-Context Learning. EMNLP 2023 Best Paper.
- Understanding In-Context Learning in Transformers and LLMs by Learning to Learn Discrete Functions. ICLR 2024.

2. It would be better to provide controlled experiments or case studies demonstrating how different forms of ICL (e.g., few-shot examples versus instructions) behave under a unified metric. Meanwhile, it was highly recommended to offer a more formal or quantitative framework that distinguishes between trivial context effects and genuine meta-learned adaptations.

**Strengths And Weaknesses:**

1. The submission contains no original experiments or quantitative results. All claims remain speculative without any new empirical validation.

2. The proposed expanded definition of ICL does not advance a new theory. The paper largely repackages existing ideas without rigorous formal analysis. The contributions of expanding ICL concepts based on existing works with pure theoretical claims can be weak, providing limited insights for future research.

3. The paper’s scope is very wide, covering many instances of context-dependent behavior. As a result, it lacks focus and fails to demonstrate a clear advantage over existing survey and perspective work.

4. Several key papers analyzing ICL are missing (See Requested Changes).

---

> ### Author Response · Authors · 2025-05-02
>
> We thank the reviewer for their thoughtful review.
>
> * **Goal:** Our goal in this paper was primarily to write a survey article highlighting these connections between different types of meta-learning — indeed, TMLR accepts survey articles as long as they draw new connections or propose new problems. Although as the reviewer notes we are building on prior literature in our work, we believe that the strength of our survey is precisely in its breadth: making more explicit the links between topics in natural language processing and meta-learning that are typically studied separately, between topics in linguistics and in-context learning that are usually seen as distinct, and and highlighting important issues, including the importance of considering interactions between different types of in-context learning, and different modes of generalization.
> * **Supplemental experiment:** Our goal in this paper was primarily to write a survey article highlighting these connections between different types of meta-learning. We see our main contribution as drawing out these common themes from prior work, emphasizing their connections, and providing a unifying perspective along with suggestions for future research, rather than performing our own experiments. However, as several reviewers requested it, we have added some minimal experiments to the supplement that illustrate some of the themes we raise, by showing comparisons and interactions between different types of learning. In particular, we compare patterns of responses produced by a language model within and across different types of prompts (instructions, few-shot examples, and examples with explanations). We find several interesting and statistically significant patterns: 1) different prompts of the same type produce significantly more similar patterns of responses than responses of different types, 2) specifically, different instruction prompts and different few-shot prompts produce response patterns that are more similar to other prompts of the same type than prompts of the other type, 3) explanations added after each few-shot example increase the consistency of response patterns among few-shot prompts. These results support our claims that different modes of in-context learning yield different modes of adaptation to a task, and that these can interact in complex ways. In our view, these results strengthen our argument for creating a broad unifying framework that highlights the similarities, differences, and interactions among different kinds of contextual adaptation.
> * **References:** We appreciate the reviewer pointing out these related works, and have added  references to them.

---

### Review · Reviewer_3pWA · 2025-05-01

**Summary Of Contributions:**

The paper presents a broad perspective on in-context learning (ICL), arguing that any sequence task where context reduces prediction loss can be seen as a form of ICL. There are not only the traditional few-shot supervised form, but also instruction following, role-playing, and time-series extrapolation. These are uniformed in the natural language understanding capacity of LLMs.

**Audience:**

No

**Claims And Evidence:**

No

**Requested Changes:**

1. A clear definition of ICL with formulation, what is included and excluded is necessary. It also need to clarified what is defined in literature and what is new.

2. I would kindly suggest the authors to publish this as a popular science article rather than an academic paper.

**Strengths And Weaknesses:**

Strength: 1. This paper is easy to follow.
2. It reviews a wide range of literature related to LLM and meta-learning.

Weakness:
1. The way that this paper extends the definition of ICL is not reasonable enough. The oracle definition of ICL was refers to the few-shot supervised form [1], to distinguish the capacity of learning from examples. This paper extends it far too general: it seems that any feeding-forward process with informative context is ICL. Such a general definition is meaningless that hard to exclude anything.

2. There lacks meaningful contribution. Most information provided in this paper are already well-known. There is neither novel understandings nor techniques.

[1] Brown T, Mann B, Ryder N, et al. Language models are few-shot learners[J]. Advances in neural information processing systems, 2020, 33: 1877-1901.

---

> ### Author Response · Authors · 2025-05-02
>
> **Survey:**
> One category of article that TMLR accepts (per https://jmlr.org/tmlr/editorial-policies.html) is “surveys that draw new connections, highlight trends, and suggest new problems in an area.” Our goals in this work were to provide a survey of existing works that both draws new connections and highlights new problems in an area. We draw connections for example between 1) standard few-shot supervised in-context learning and other types of contextual adaptation behaviors in language models, 2) the various types of adaptation seen in the meta-learning literature, 3) simpler linguistic phenomena that have similar contextual adaptation patterns. We also highlight new problems, including 1) considering behavioral and mechanistic interactions between different types of in-context learning, and 2) exploring different types of generalization when studying in-context learning. Thus, from our perspective, this paper clearly meets the standards of survey publication in TMLR.
>
> **Definition:**
> * In fact, as we note in the paper, [1] also included instructions (rather than just few-shot examples) within their definition of in-context learning. Moreover, as we note, the notion of measuring in-context learning through the reduction in loss on later tokens was already established in seminal works on scaling laws and induction heads: “Indeed, some early measures of in-context learning in LMs effectively capture this broader spectrum of in-context adaptation. For example, Kaplan et al. (2020) and later Olsson et al. 2022 measured ICL by the lower loss for tokens later in a document relative to earlier tokens — which incorporates all the various contextual structures that support that loss reduction.” Finally, this perspective is supported by earlier work in the meta-learning literature (Ortega et al., 2019). We therefore believe part of the value of our work is highlighting this notion, and the many types of contextual structures that contribute to loss reduction, more explicitly — and linking them to the prior meta-learning literature, as noted above.
> * We do wish to note that our definition already incorporates two important restrictions on the breadth of ICL: first, that the loss reduction must be present in expectation over the data distribution, and second that it must be non-trivial (i.e. better than the distribution conditioned on the timestep alone; see footnote 2 for an example of a type of task excluded by this restriction).
> * If the reviewer has other specific restrictions that they believe would improve our definition, while still accommodating the many ways that models can adapt in context (e.g. from instructions alone, rather than just examples, or with explanations), and the breadth of ways this in-context learning can be applied (e.g. producing code for a task than performing a task directly) — particularly a definition grounded in the prior literature — we would invite them to make a specific suggestion.
>
> **Other revisions**
> Although our paper is primarily a survey, at the suggestion of the other reviewers we have added some brief experiments to the supplement showing comparisons and interactions between different types of learning in response to suggestions from other reviewers. In our view, these results strengthen our argument for creating a broad unifying framework that highlights the similarities, differences, and interactions among different kinds of contextual adaptation.

---

### Decision · Action_Editor_nvE1 · 2025-05-21

**Recommendation:** Reject

**Comment:**

The reviewers acknowledge the clarity of writing and appreciate the effort to unify disparate threads of ICL research under a broader meta-learning framework. However, several reviewers note that the work lacks sufficient rigor, or depth to merit publication in its current form. The conceptual idea are not accompanied by formal analysis, or empirical evaluation that would elevate the claims beyond synthesis. Furthermore, some concerns were raised about ambiguous definitions and the generality of assertions made without empirical backing. While the perspective is potentially useful, it would benefit from more concrete demonstrations or theoretical development to support the proposed framework.

**Audience:**

The paper may interest researchers who think about how language models adapt to different tasks or who study meta-learning. However, because the paper does not include clear methods, experiments, or formal analysis, it will be hard for most readers to use or build on its ideas.

**Claims And Evidence:**

This paper offers a broad view of in-context learning (ICL), suggesting that many language model behaviors, such as few-shot learning, instruction following, and coreference resolution, can be seen as different types of ICL. While this is an interesting idea, the paper does not provide enough evidence to support it. There are no formal definitions or new experiments. Most of the arguments are based on examples or summaries of existing work. As a result, the claims remain speculative and are not strongly supported.